# Preserving Pre-trained Features Helps Calibrate Fine-tuned Language Models

**Guande He**[1], **Jianfei Chen**[1*], **Jun Zhu**[1,2*]
[1]Dept. of Comp. Sci. & Tech., Institute for AI, Tsinghua-Bosch Joint Center for ML,
BNRist Center, State Key Lab for Intell. Tech. & Sys., Tsinghua University, Beijing, China
[2]Pazhou Lab, Guangzhou, 510330, China
`heguande17@outlook.com, {jianfeic, dcszj}@tsinghua.edu.cn`

## Abstract

Large pre-trained language models (PLMs) have demonstrated strong performance on natural language understanding (NLU) tasks through fine-tuning. However, fine-tuned models still suffer from overconfident predictions, especially in out-of-domain settings. In this paper, we tackle the problem of calibrating fine-tuned language models. We demonstrate that the PLMs are well-calibrated on the masked language modeling task with robust predictive confidence under domain shift, yet the fine-tuned models fail to retain such property due to catastrophic forgetting, which impacts the calibration on the downstream classification task. In light of these observations, we evaluate the calibration of several methods that preserve pre-trained features and show that preserving pre-trained features can improve the calibration of fine-tuned language models. Among these methods, our proposed method that encourages the fine-tuned model to learn generative representations with auxiliary language modeling objective achieves competitive accuracy and the lowest expected calibration error compared to several strong baselines under both in-domain and out-of-domain settings on three downstream NLU tasks.

## 1 Introduction

Fine-tuning pre-trained language models (PLMs) is a dominating paradigm for natural language understanding (NLU) with state-of-the-art results for a variety of NLU tasks (Peters et al., 2018; Devlin et al., 2019; Liu et al., 2019; He et al., 2021a). The powerful fine-tuned language models have been experimented with for decision-making in real-world applications such as the healthcare domain (He et al., 2020) and safety-critical domain (Sandagiri et al., 2020), where the classification networks need to be highly accurate and provide calibrated confidence for their predictions to improve the safety and trustiness of the models (Guo et al., 2017). For example, suppose a medical language inference LM that predicts the disease given the description of symptoms is well-calibrated, i.e., the model's posterior probabilities (or confidence) align well with the true correctness likelihood. In that case, the wrong predictions can be easier to detect and correct by human doctors by given low predictive confidence. However, as with other modern neural networks, the fine-tuned LMs are shown to suffer from overconfidence (Desai & Durrett, 2020; Jiang et al., 2021), which creates obstacles and concerns for their deployment in real-world applications.

Uncertainty estimation of fine-tuned models is challenging due to the small amount of available data for fine-tuning, especially under out-of-domain settings (Desai & Durrett, 2020; Guo et al., 2021). While prior works illustrate that simple calibration techniques such as temperature scaling (Guo et al., 2017) and label smoothing (Szegedy et al., 2016) are not sufficient to calibrate the fine-tuned LMs under both in-domain (ID) and out-of-domain (OD) settings (Desai & Durrett, 2020; Park & Caragea, 2022), several approaches with strong regularization have been developed to calibrate the fine-tuned model on NLU tasks, including knowledge distillation from deep ensembles (Guo et al., 2021), stochastic network architectures (Fan et al., 2020; Zhang et al., 2021), and Mixup (Park & Caragea, 2022). However, these existing works mostly utilize general calibration methods for

---

*Corresponding authors.

supervised learning, while specific properties of the pre-training & fine-tuning paradigm are still largely neglected.

In this work, we tackle the calibration of fine-tuned models from the perspective of better leveraging the powerful PLMs. Through a carefully designed empirical study on both pre-trained and fine-tuned models, we first observe that PLMs themselves are actually *well-calibrated* on the masked language modeling (MLM) task and robust to higher levels of perturbation to the inputs, which suggests the PLMs can model the predictive uncertainty well across different domains. However, the pre-trained features are only used as initialization and are distorted by the fully discriminative fine-tuning. The phenomenon is known as catastrophic forgetting (McCloskey & Cohen, 1989; Kirkpatrick et al., 2017; Howard & Ruder, 2018). We show that such forgetting can make the fine-tuned language models fail to hold proper predictive confidence toward the OD and outlier samples, which leads to miscalibration on the downstream tasks. Based on the observations, we hypothesize that preserving the pre-trained features helps calibrate the fine-tuned LMs.

To validate our hypothesis, we first evaluate the calibration of some previous methods that can preserve pre-trained features, including (1) Parameter-efficient tuning (Houlsby et al., 2019; Hu et al., 2021; Li & Liang, 2021), (2) Pre-trained weight decay, (3) Mixout (Lee et al., 2020). Although these methods were originally designed to improve the performance beyond uncertainty estimation, our experiment demonstrates that these methods outperform vanilla fine-tuning in terms of calibration, especially under out-of-domain settings. Based on our observation that the PLMs are well-calibrated on the MLM task yet the fine-tuned LMs that forget the pre-trained features struggle with overconfidence under domain shift, we propose a simple baseline that utilizes the MLM objective to maintain the consistency between the pre-trained and fine-tuned models. The proposed method achieves the lowest expected calibration error and competitive accuracy compared to existing calibration methods in both ID and OD settings on three NLU tasks, including natural language inference, paraphrase detection, and commonsense reasoning, showing that preserving the pre-trained features is an effective approach for improving the calibration of fine-tuned LMs.

## 2 PRELIMINARIES

### 2.1 MASKED LANGUAGE MODELS

Masked language models generally consist of a transformer-based text encoder $f_\phi$ parameterized by $\phi$ and a linear language modeling head $g_\theta$ parameterized by $\theta$. In the pre-training phase, the model handles the masked language modeling task (Devlin et al., 2019). Assume we have unsupervised sequence inputs $\boldsymbol{x}$ sampled from large-scale corpora $p_u(\boldsymbol{x})$. A subset of $\boldsymbol{x}$ is first masked by a corruption function or distribution. Denote the indices of the masked tokens as $\mathbb{M}$, the set of masked tokens as $\boldsymbol{x}_{\mathbb{M}}$, and the observed unmasked input as $\boldsymbol{x}_{\backslash\mathbb{M}}$. The model is trained to recover the masked tokens $\boldsymbol{x}_{\mathbb{M}}$. In particular, the masked language model first uses the text encoder to get a hidden representation of the input, denoted as $f_\phi(\boldsymbol{x}_{\backslash\mathbb{M}})$. Then the language modeling head $g_\theta$ with softmax function is applied to $f_\phi(\boldsymbol{x}_{\backslash\mathbb{M}})$ to obtain a conditional categorical distribution $p_{\mathrm{mlm}}(x_i|\boldsymbol{x}_{\backslash\mathbb{M}})$ over the vocabulary $\mathbb{V}$ for each masked position $i \in \mathbb{M}$. The masked language modeling objective is:

$$\mathcal{L}_{\mathrm{mlm}} = -\mathbb{E}_{p_u(\boldsymbol{x})}\big[\sum_{i\in\mathbb{M}} \log\, p_{\mathrm{mlm}}(x_i|\boldsymbol{x}_{\backslash\mathbb{M}}; \phi, \theta)\big] \tag{1}$$

In the fine-tuning phase, assume we have labeled data in the form of $(\boldsymbol{x}, y)$ sampled from data distribution $p_d$, where $\boldsymbol{x}$ corresponds to the text input, and $y$ corresponds to the label. For classification tasks, a task-specific head $h_\varphi$ that parameterized by $\varphi$ is applied on the hidden representation of input to obtain the logit for each class. The predictive posterior distribution $q(y|\boldsymbol{x})$ is given by the logits after softmax operation. In standard fine-tuning, the pre-trained encoder $f_\phi$ and the task-specific head $h_\varphi$ are jointly optimized using the cross-entropy loss:

$$\mathcal{L}_{\mathrm{cls}} = -\mathbb{E}_{p_d(\boldsymbol{x},y)}\left[\log\, q(y|\boldsymbol{x}; \phi, \varphi)\right] \tag{2}$$

which is also known as full fine-tuning (Full-FT) (Peters et al., 2018; Devlin et al., 2019).

## 2.2 CONFIDENCE CALIBRATION

The framework of confidence calibration under the supervised classification setting can be expressed as a joint distribution $P(\hat{y}, \hat{p})$ over the label prediction $\hat{y} \in |\mathcal{Y}|$ and the corresponding confidence $\hat{p} \in [0, 1]$. A perfectly calibrated model holds $P(\hat{y} = y | \hat{p} = p) = p$ (Guo et al., 2017). One way to evaluate calibration through finite samples is *expected calibration error*, i.e., ECE (Naeini et al., 2015). To compute ECE, the model's predictive confidences are first grouped into $M$ equal-sized bins. Denote $B_m$ as the indices of samples whose confidences are in the interval $(\frac{m-1}{M}, \frac{m}{M}]$. Suppose we have $N$ samples, the ECE is calculated by the weighted average of the difference between confidence and accuracy in each bin:

$$\text{acc}(B_m) = \frac{1}{|B_m|} \sum_{i \in B_m} \mathbf{1}\left(\hat{y}_i = y_i\right), \ \ \text{conf}(B_m) = \frac{1}{|B_m|} \sum_{i \in B_m} \hat{p}_i$$

$$\text{ECE} = \sum_{m=1}^{M} \frac{|B_m|}{N} |\text{acc}(B_m) - \text{conf}(B_m)| \tag{3}$$

In this work, we set $M = 10$ following Desai & Durrett (2020).

## 3 A CLOSER LOOK TO THE PRE-TRAINED AND FINE-TUNED LANGUAGE MODELS IN CALIBRATION

In this section, we explore the connection between the pre-trained and fine-tuned language models in terms of calibration by examining: (1) The calibration of the pre-trained language models themselves. (2) How fine-tuning affects calibration on the downstream classification tasks.

### 3.1 WHAT WE FORGET AFTER FINE-TUNING THE PRE-TRAINED LANGUAGE MODELS

Pre-trained language models have demonstrated their ability to capture informative linguistic features from large corpora (Tenney et al., 2019; Jawahar et al., 2019; Ethayarajh, 2019). Intuitively, the pre-trained features learned on diverse corpora should be capable of performing uncertainty estimation well. In this subsection, we validate that the pre-trained language models are indeed well-calibrated on the MLM task, which suggests that the predominant full fine-tuning method that forgets such pre-trained features is suboptimal.

Table 1: The Expected Calibration Error (ECE) of the pre-trained RoBERTa$_{\text{BASE}}$ on the MLM task.

| Dataset | $p_{\text{mask}} = 0.15$ | $p_{\text{mask}} = 0.3$ | $p_{\text{mask}} = 0.5$ |
|---|---|---|---|
| WikiText-103 | $3.89_{0.23}$ | $3.73_{0.12}$ | $4.30_{0.08}$ |
| SNLI | $2.99_{0.29}$ | $3.65_{0.18}$ | $5.80_{0.18}$ |
| MNLI | $2.68_{0.29}$ | $3.61_{0.10}$ | $5.65_{0.18}$ |
| QQP | $4.23_{0.13}$ | $4.74_{0.06}$ | $6.19_{0.05}$ |
| TwitterPPDB | $7.52_{0.23}$ | $8.45_{0.07}$ | $10.12_{0.10}$ |
| SWAG | $5.67_{0.08}$ | $5.86_{0.02}$ | $6.88_{0.05}$ |
| HellaSWAG | $3.18_{0.10}$ | $3.44_{0.03}$ | $4.82_{0.02}$ |

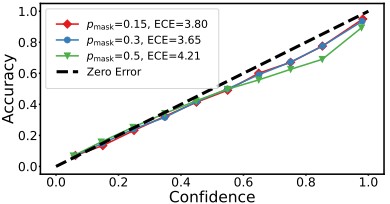

Figure 1: Reliability diagram of the pre-trained RoBERTa$_{\text{BASE}}$ on the MLM task on WikiText-103.

**Setup:** We evaluate the calibration of the pre-trained RoBERTa$_{\text{BASE}}$ (Liu et al., 2019) through the MLM task. We use the test split of the WikiText-103 (Merity et al., 2016) , one of the corpora in the pre-training phase, and six downstream datasets from different domains (see §5.1 and A.1 for details) as sequence inputs for masked language modeling. The inputs are masked and corrupted with three levels of 15%, 30%, and 50% mask probability with the same masking approach (i.e., the 80-10-10 strategy) as in the pre-training phase (Devlin et al., 2019).

**Results and Analysis:** Table 1 and Figure 1 show the ECE and the reliability diagram (DeGroot & Fienberg, 1983; Niculescu-Mizil & Caruana, 2005) of the pre-trained RoBERTa$_{\text{BASE}}$ on the MLM task. The results suggest that the PLM is relatively well-calibrated across different domains, where the model needs to recover the corrupted position with the options of $|\mathbb{V}|$. Moreover, as the mask

probability grows higher than in the pre-training phase, the ECE of the PLM increases only by a relatively small amount, which indicates that the PLM's predictive confidence $p_{\mathrm{mlm}}(x|\boldsymbol{x}_{\backslash\mathbb{M}})$ on the MLM task is robust to higher corruption levels. Figure 2 demonstrates that although the hidden representations of 50% masked inputs (visualized by t-SNE (Van der Maaten & Hinton, 2008)) have shifted significantly from the original input, the PLM can still make calibrated predictions to the original inputs. Intuitively, the calibrated confidence on the MLM task suggests that the pre-trained features of PLMs are good at modeling the samples under large domain shifts, which may benefit the calibration of the downstream classification task under OD settings. However, this property is less likely to be retained by the fine-tuned LM due to catastrophic forgetting caused by full fine-tuning with the discriminative objective (Howard & Ruder, 2018).

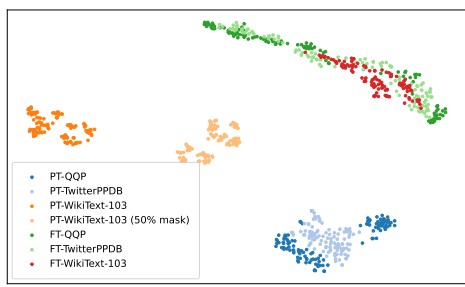

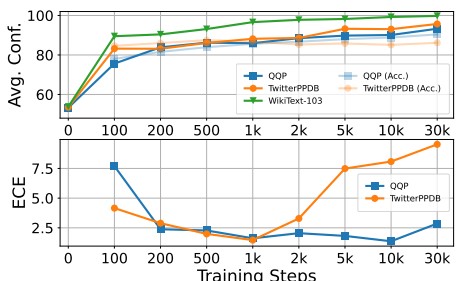

Figure 2: t-SNE visualization for hidden representations of the sampled inputs from different domains given by the pre-trained (PT) and fine-tuned (FT) language models.

Figure 3: Average predictive confidence (up) and ECE (down) for the validation split of QQP (ID), TwitterPPDB (OD), and WikiText-103 (outlier) dataset in different training steps.

## 3.2 HOW FINE-TUNING AFFECTS CALIBRATION ON THE DOWNSTREAM TASKS

Although previous works have shown that fine-tuned language models can outperform non-pre-trained models in terms of calibration (Desai & Durrett, 2020), the fine-tuned LMs' calibration performance is still far from satisfactory, especially under the OD settings (Desai & Durrett, 2020; Guo et al., 2021). To study why fine-tuned LMs are miscalibrated under the OD settings, we conduct a case study on the LM fine-tuned on the QQP dataset, which is a typical failure case that the fine-tuned model exhibits dramatic disparity in ECE on ID and OD settings, as shown in Figure 3.

**Setup:** We fine-tune the pre-trained RoBERTa$_{\mathrm{BASE}}$ on the QQP training set following the default configuration of the Huggingface Transformers library (Wolf et al., 2020). Compared to the pre-trained model, we visualize the hidden representations of the same inputs with §3.1 given by the fine-tuned model. Besides, we evaluate the average confidence of the fine-tuned model's prediction on several datasets, including the in-domain QQP validation set, the out-of-domain TwitterPPDB validation set, and the outlier WikiText-103 validation set that does not hold any particular attributes of the downstream classification task.

**Results and Analysis:** As shown in Figure 2, compared to the pre-trained models, fine-tuning changes the hidden representation of the LM in two ways: (1) For the inputs within the same domain, fine-tuning enlarges the difference of the corresponding hidden representations, which aligns with the quantitative results of the previous work (Zhou & Srikumar, 2022). (2) For the inputs across different domains, fine-tuning makes the hidden representations from different domains much harder to distinguish by projecting them to a simpler data manifold, which causes the fine-tuned model fails to give proper predictive confidence for OD and outlier samples.

As shown in Figure 3, the average predictive confidence of the ID, OD, and outlier validation sets increases as the training step increases. The gap between the average predictive confidence and the correctness likelihood (i.e., classification accuracy) in the OD setting is relatively larger than in the ID setting, which results in a larger OD ECE. More crucially, the average confidence of the outlier samples is higher during the whole training process than both ID and OD settings and increases to nearly 100% after three training epochs, which can not be fixed by simple techniques such as temperature scaling and early stopping. Ideally, the model should be uncertain about the outlier samples significantly deviating from the training samples. However, the fine-tuned LM exhibits overcon-

fidence toward the OD and outlier samples, which implies that strong regularization methods are needed to improve the confidence modeling for OD and outlier samples. Based on the observations that the pre-trained features of the PLMs can model the predictive uncertainty well across different domains and are distorted by the fine-tuned LMs in §3.1, we hypothesize that *preserving the pre-trained features of PLMs helps the fine-tuned LMs better model the predictive confidence and improve calibration on downstream classification tasks.*

## 4 METHODS

To validate the hypothesis that preserving the pre-trained features helps the calibration of fine-tuned LMs, we examine existing methods that can preserve the pre-trained features in different ways. Although these methods are not originally designed for enhancing uncertainty estimation, such as achieving better trade-off between tunable parameters and model's performance for parameter-efficient tuning, or improving classification accuracy and stability for pre-trained weight decay and Mixout, we anticipate that these methods may improve calibration by mitigating catastrophic forgetting and evaluate their effectiveness in calibration in §5.

### 4.1 PARAMATER-EFFICIENT TUNING

Parameter-efficient tuning is a series of fine-tuning methods which keep the pre-trained parameters of the text encoder $\phi$ frozen and update only a small number of extra parameters $\phi_\Delta$ and the task-specific head $h_\varphi$ while preserving competitive performance with Full-FT. Since the pre-trained knowledge is encoded to the model's parameters and there are only a small amount of extra parameters, parameter-efficient tuning methods can preserve more pre-trained features compared to Full-FT. In this work, we choose three mainstream parameter efficient tuning methods: (1) **Adapter** (Houlsby et al., 2019), which adds a light-weight bottleneck module after the output of each sub-layer in the transformer block; (2) **LoRA** (Hu et al., 2021), which updates the attention weight matrix using low-rank reparameterization; (3) **Prefix Tuning** (Li & Liang, 2021), which prepends tunable prefix vectors to keys and values of the multi-head attention layers.

### 4.2 REGULARIZATION WITH PRE-TRAINED WEIGHT

Introducing regularization terms using the pre-trained weight can also better leverage the pre-trained features during fine-tuning. In this work, we adopt two common regularization techniques:

**Pre-trained Weight Decay:** Traditional weight decay methods add a regularization term $\frac{\lambda}{2}||\boldsymbol{w}||^2$ that penalizes large weights to improve generalization (Krogh & Hertz, 1991), where $\lambda$ is a regularization coefficient. As an alternative, performing weight decay towards the pre-trained weight $\boldsymbol{w}_0$ by adding $\frac{\lambda}{2}||\boldsymbol{w} - \boldsymbol{w}_0||^2$ to the task loss function is shown by previous works as an effective way to mitigate catastrophic forgetting caused by fine-tuning (Wiese et al., 2017) and can improve the performance of the downstream task (Chen et al., 2020).

**Mixout:** To explicitly prevent the deviation from the pre-trained weight $\boldsymbol{w}_0$, Lee et al. (2020) propose Mixout that stochastically replaces the model parameters with their pre-trained counterparts with probability $p$ at each training iteration, which has been shown to improve the stability of fine-tuning and enhance the classification accuracy of the fine-tuned LMs on downstream task.

### 4.3 JOINT LEARNING WITH MLM OBJECTIVE

Besides fixing or constraining the model parameters to the pre-trained counterpart, we can enforce the consistency between the pre-trained and fine-tuned LMs by utilizing the MLM objective. Previous works have demonstrated that performing the MLM task before or during the fine-tuning process can yield better performance on the downstream tasks (Sun et al., 2019; Wiedemann et al., 2020; Ma et al., 2021). In this work, to better preserve the pre-trained features, we jointly optimize the MLM objective in the fine-tuning phase: $\mathcal{L}_{\text{joint}} = \alpha_{\text{mlm}} \mathcal{L}_{\text{mlm}} + \mathcal{L}_{\text{cls}}$, where $\alpha_{\text{mlm}}$ is the scaling factor of the MLM loss. In addition to introducing the MLM objective, we propose three simple techniques to further strengthen the connection between the pre-trained model and our fine-tuned model:

**Utilizing corpus of the pre-training phase:** The dataset for the MLM task is not required to be labeled as for the downstream task, so we can choose any text dataset. In this work, we present two criteria for selecting $p_u(\boldsymbol{x})$ of the MLM task. The first one uses the dataset of downstream tasks, referred to as JL-D. The second one introduces the corpus of the pre-training phase, referred to as JL-P. We suppose that using the corpus in the pre-training phase is helpful in preserving the pre-trained features, while increasing data diversity could enhance the model's uncertainty estimation under OD settings.

**Distillation from the pre-trained model:** Knowledge distillation (Hinton et al., 2015) has been shown to be an effective technique to reap the benefits of a powerful teacher model. In addition to preserving the accuracy, Guo et al. (2021) illustrate that the calibration performance of the teacher model can be distilled into the student model and propose to distill from an ensemble of fine-tuned LMs for better calibration. Inspired by this, we perform knowledge distillation from the pre-trained language model when performing the MLM task. Specifically, instead of calculating the MLM loss using the original text as hard labels, we use the KL-divergence $D_{\mathrm{KL}}(p_{\mathrm{mlm}}(x_i|\boldsymbol{x}_{\backslash\mathbb{M}}; \phi_0, \theta_0)\|p_{\mathrm{mlm}}(x_i|\boldsymbol{x}_{\backslash\mathbb{M}}; \phi, \theta))$ between the predictive distribution $p_{\mathrm{mlm}}(x_i|\boldsymbol{x}_{\backslash\mathbb{M}}; \phi_0, \theta_0)$ of the pre-trained language model and our model.

**Regularization on the contextualized representation:** Zhou & Srikumar (2022) validate that fine-tuning enlarges the distance in feature space between samples from different classes. This behavior may increase the deviation of the fine-tuned model from the pre-trained model. To address this problem, we introduce a heuristic regularization by adding the $L^2$ norm of each training example's contextualized representation $f_\phi(\boldsymbol{x})$ with regularization coefficient $\beta_{L^2}$.

Putting $\mathcal{L}_{\mathrm{joint}}$ with the knowledge distillation and regularization term together, we get our final joint learning objective:

$$
\begin{aligned}
-\mathcal{L}_{\mathrm{JL}} =\; & \alpha_{\mathrm{mlm}} \, \mathbb{E}_{p_u(\boldsymbol{x})} \big[ \sum_{i \in \mathbb{M}} D_{\mathrm{KL}}(p_{\mathrm{mlm}}(x_i|\boldsymbol{x}_{\backslash\mathbb{M}}; \phi_0, \theta_0) \| p_{\mathrm{mlm}}(x_i|\boldsymbol{x}_{\backslash\mathbb{M}}; \phi, \theta)) \big] \\
& + \mathbb{E}_{p_d(\boldsymbol{x}, y)} \left[ \log q(y|\boldsymbol{x}; \phi, \varphi) \right] + \beta_{L^2} \| f_\phi(\boldsymbol{x}) \|
\end{aligned}
\tag{4}
$$

Following previous works (Desai & Durrett, 2020; Park & Caragea, 2022), we also apply label smoothing (Szegedy et al., 2016) on the classification task, which can mitigate overconfident predictions by distributing a $\sigma$ fraction of probability mass of the ground-truth label equally to other non-ground-truth classes.

## 5 EXPERIMENTS

### 5.1 GENERAL SETUP

**Datasets:** We conduct experiments on three natural language understanding tasks: natural language inference (NLI), paraphrase detection (PD), and commonsense reasoning (CR). Each task consists of a pair of in-domain (ID) and out-of-domain (OD) datasets. Specifically, SNLI (Bowman et al., 2015) and MNLI (Williams et al., 2018) are ID and OD datasets for NLI; QQP (Shankar et al., 2017) and TwitterPPDB (Lan et al., 2017) are ID and OD datasets for PD; SWAG (Zellers et al., 2018) and HellaSWAG (Zellers et al., 2019) are ID and OD datasets for CR. We use the same train/validation/test split for those six datasets published by Desai & Durrett (2020). For each task, we fine-tune the model using the ID training set and evaluate the model's performance with both ID and OD test sets. We use WikiText-103 (Merity et al., 2016) as the corpus of the pre-training phase for JL-P. The detailed statistics for each dataset can be found in Appendix A.1.

**Setup:** We follow the general training configuration provided by the Huggingface Transformers library (Wolf et al., 2020). For parameter-efficient tuning methods, we use the default hyperparameter configuration provided by OpenDelta (Ding et al., 2022) library for all three methods and conduct a grid search for learning rates on different tasks. For pre-trained weight decay (PWD), we follow the implementation of the RecAdam (Chen et al., 2020), which integrates the quadratic penalty between the model parameters and the pre-trained parameters into the Adam optimizer (Kingma & Ba, 2015). We tune the regularization strength $\lambda_{\mathrm{PWD}}$ for each task. For Mixout, we tune the mixout

probability $p_{\text{mixout}}$. For joint learning methods, we use a Bernoulli corruption distribution [1]. We conduct hyperparameter search for the mask probability $p_{\text{mask}}$, the scaling factor $\alpha_{\text{mlm}}$ of the MLM loss, and the regularization coefficient $\beta_{L^2}$ on the contextualized representation. We also tune the hyperparameter $\sigma_{\text{ls}}$ for label smoothing (LS). We search all the hyperparameters on the validation set of each task independently. Completed setup details for each method on each task can be found in Appendix A.2. All experiments are run for 3 training epochs and are deployed on a single NVIDIA A40 48G GPU within 3 hours to fine-tune a single model.

**Evaluation:** In this section, we use pre-trained `RoBERTa`$_{\text{BASE}}$ (Liu et al., 2019) for all experiments. For each NLU task, we report accuracy and expected calibration error (ECE) on both ID and OD test sets. We evaluate "out-of-the-box" calibration of our method, which does not apply any post-hoc calibration methods such as temperature scaling (Guo et al., 2017).

Table 2: Out-of-the-box calibration results of different fine-tuning methods on in-domain (SNLI, QQP, SWAG) and out-of-domain (MNLI, TwitterPPDB, HellaSWAG) datasets. We report the averaged accuracy and ECE across five random fine-tuning runs. We also report the corresponding standard deviation in subscripts.

| RoBERTa-base | In-Domain | | Out-of-Domain | |
|---|---|---|---|---|
| | Acc | ECE | Acc | ECE |
| **Task: SNLI/MNLI** | | | | |
| Baseline (Desai & Durrett, 2020) | 91.23 | 1.93 | 78.79 | 3.62 |
| Baseline (Our run) | $91.89_{0.18}$ | $2.21_{0.12}$ | $79.87_{0.32}$ | $4.06_{0.20}$ |
| BABN (Zhang et al., 2021) | 91.70 | 2.62 | 79.86 | 2.67 |
| Mixup (Park & Caragea, 2022) | $91.24_{0.3}$ | $1.28_{0.6}$ | $78.86_{0.5}$ | $1.37_{1.7}$ |
| Adapter (Houlsby et al., 2019) | $91.30_{0.09}$ | $1.32_{0.20}$ | $79.17_{0.30}$ | $1.98_{0.47}$ |
| LoRA (Hu et al., 2021) | $89.86_{0.23}$ | $1.28_{0.12}$ | $77.52_{0.23}$ | $1.70_{0.33}$ |
| Prefix Tuning (Li & Liang, 2021) | $89.08_{0.19}$ | $1.68_{0.08}$ | $76.09_{0.28}$ | $1.53_{0.39}$ |
| Pre-trained Weight Decay | $91.09_{0.16}$ | $1.71_{0.15}$ | $78.35_{0.30}$ | $\mathbf{0.87}_{0.32}$ |
| Mixout (Lee et al., 2020) | $90.70_{0.07}$ | $1.70_{0.07}$ | $78.82_{0.26}$ | $1.14_{0.39}$ |
| JL-D (w/o KD) | $\mathbf{91.95}_{0.12}$ | $\mathbf{0.67}_{0.05}$ | $79.85_{0.29}$ | $1.26_{0.50}$ |
| JL-P (w/ KD) | $91.74_{0.15}$ | $1.09_{0.15}$ | $79.36_{0.27}$ | $1.20_{0.43}$ |
| JL-P (w/ KD) + LS | $91.78_{0.09}$ | $1.48_{0.17}$ | $\mathbf{80.00}_{0.15}$ | $1.90_{0.07}$ |
| **Task: QQP/TwitterPPDB** | | | | |
| Baseline (Desai & Durrett, 2020) | 91.11 | 2.33 | 86.72 | 9.55 |
| Baseline (Our run) | $91.26_{0.13}$ | $2.44_{0.10}$ | $86.15_{0.35}$ | $10.09_{0.49}$ |
| BABN (Zhang et al., 2021) | $\mathbf{91.72}$ | 1.74 | 87.31 | 9.42 |
| Mixup (Park & Caragea, 2022) | $89.75_{0.6}$ | $2.18_{0.7}$ | $\mathbf{87.63}_{1.0}$ | $3.96_{1.6}$ |
| Adapter (Houlsby et al., 2019) | $90.18_{0.09}$ | $1.37_{0.08}$ | $85.63_{0.30}$ | $9.84_{0.47}$ |
| LoRA (Hu et al., 2021) | $88.54_{0.10}$ | $1.05_{0.09}$ | $85.51_{0.46}$ | $8.55_{0.51}$ |
| Prefix Tuning (Li & Liang, 2021) | $87.67_{0.12}$ | $1.46_{0.16}$ | $85.26_{0.22}$ | $8.02_{0.21}$ |
| Pre-trained Weight Decay | $89.95_{0.09}$ | $1.25_{0.09}$ | $85.28_{0.29}$ | $8.01_{0.35}$ |
| Mixout (Lee et al., 2020) | $89.77_{0.08}$ | $1.17_{0.09}$ | $83.69_{1.72}$ | $5.75_{1.24}$ |
| JL-D (w/o KD) | $90.42_{0.08}$ | $\mathbf{0.57}_{0.12}$ | $86.67_{0.22}$ | $6.49_{0.44}$ |
| JL-P (w/ KD) | $90.50_{0.09}$ | $0.91_{0.50}$ | $86.50_{0.90}$ | $\mathbf{1.27}_{0.39}$ |
| JL-P (w/ KD) + LS | $90.42_{0.08}$ | $0.77_{0.28}$ | $86.38_{0.93}$ | $2.21_{1.06}$ |
| **Task: SWAG/HellaSWAG** | | | | |
| Baseline (Desai & Durrett, 2020) | 82.45 | 1.76 | 41.68 | 11.93 |
| Baseline (Our run) | $83.98_{0.27}$ | $1.38_{0.25}$ | $42.99_{0.97}$ | $8.80_{0.65}$ |
| BABN (Zhang et al., 2021) | 83.12 | 1.32 | 43.11 | 9.72 |
| Mixup (Park & Caragea, 2022) | $82.69_{0.7}$ | $1.12_{0.4}$ | $41.37_{1.1}$ | $1.86_{0.9}$ |
| Adapter (Houlsby et al., 2019) | $82.37_{0.12}$ | $3.25_{0.19}$ | $43.94_{1.08}$ | $11.13_{0.76}$ |
| LoRA (Hu et al., 2021) | $80.01_{0.18}$ | $5.12_{0.15}$ | $43.17_{0.42}$ | $4.38_{0.99}$ |
| Prefix Tuning (Li & Liang, 2021) | $80.66_{0.20}$ | $4.20_{0.56}$ | $42.86_{0.63}$ | $5.76_{1.09}$ |
| Pre-trained Weight Decay | $84.15_{0.22}$ | $1.29_{0.15}$ | $42.73_{1.54}$ | $8.21_{0.67}$ |
| Mixout (Lee et al., 2020) | $83.17_{0.12}$ | $\mathbf{0.78}_{0.09}$ | $\mathbf{45.15}_{0.74}$ | $7.52_{0.65}$ |
| JL-D (w/o KD) | $\mathbf{84.49}_{0.24}$ | $0.91_{0.25}$ | $43.75_{0.92}$ | $7.64_{0.43}$ |
| JL-P (w/ KD) | $83.51_{0.05}$ | $1.94_{0.12}$ | $44.78_{0.22}$ | $5.23_{0.43}$ |
| JL-P (w/ KD) + LS | $83.61_{0.27}$ | $1.20_{0.29}$ | $44.08_{0.61}$ | $\mathbf{1.62}_{0.29}$ |

## 5.2 MAIN RESULTS

We summarize our results in Table 2. Our baselines include full fine-tuning and two advanced methods based on the pre-trained language models: Bayesian Attention Belief Networks (BABN) (Zhang et al., 2021) and Mixup (Park & Caragea, 2022). We copied the results of the same setting from the

---

[1] For the MLM objective in joint learning method, we only perform the mask operation instead of the 80-10-10 strategy as in the pre-training phase.

original paper. We also report our full fine-tuning runs following the default behavior of the training scripts provided by the Huggingface Transformers library. We present our experimental results for pre-trained weight decay, Mixout, JL-D (w/o KD) and JL-P (w/ KD) [2]

As shown in Table 2, full fine-tuned models generally have lower in-domain ECE compared to out-of-domain, which illustrates that the fine-tuned language models tend to be overconfident under OD settings. BABN exhibits better generalization performance than deterministic methods, but it has a limited effect on the OD calibration. Mixup significantly lowers the ECE under both ID and OD settings while preserving comparable accuracy to vanilla fine-tuning. Besides, all of the methods that preserve the pre-trained features in different ways are generally more calibrated than full fine-tuning, especially under the OD settings, which matches our expectations that pre-trained features are helpful to better model the predictive confidence for OD samples. In addition to achieving competitive performance, parameter-efficient tuning methods have advantages in calibration over full fine-tuning. Fine-tuning with regularization to pre-trained models in the parameter space also significantly improves calibration. However, Table 2 and Table 5 show that this requires maintaining a relatively high constraint strength to the pre-trained weights throughout the fine-tuning process, which leads to a large loss of raw quality in some cases, e.g., Mixout on QQP/TwitterPPDB.

Notably, the proposed joint learning (JL) methods outperform previous calibration methods in ECE across three tasks simultaneously under both ID and OD settings, which suggests that it may be more effective to encourage fine-tuned models to be consistent with pre-trained models in the function space. In addition to being well-calibrated, JL models achieve the best accuracy in both ID and OD settings on NLI and CR tasks and maintain accuracy within $< 1\%$ drop compared to vanilla fine-tuning on the PD task. We also highlight that the JL models have relatively low standard deviations for both accuracy and ECE compared to other methods.

## 5.3 PRESERVING PRE-TRAINED FEATURES HELPS CALIBRATE FINE-TUNED LMS

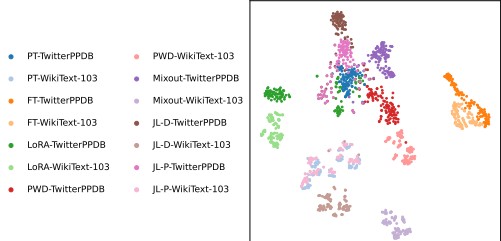

Figure 4: t-SNE visualization for hidden representations of the sampled inputs from different domains given by different models fine-tuned on QQP.

Figure 5: Confidence histogram for OD samples (up) and outlier samples (down) of different models fine-tuned on QQP.

As shown in Figure 4, compared to full fine-tuning, the hidden representations for OD samples of the methods described in §4 are more consistent with the PLMs, showing that they are able to preserve the pre-trained features and can mitigate catastrophic forgetting. The representations for outlier samples are more distinguishable from the OD samples, as the pre-trained model does. Figure 5 illustrates that preserving the pre-trained features helps calibrate the fine-tuned models by mitigating the overconfident tendency to the OD and outlier samples discussed in §3.2. Specifically, parameter-efficient tuning, pre-trained weight decay, and JL-D slightly alleviate the overconfidence toward the OD and outlier samples, while JL-P and Mixout significantly improve the fine-tuned models' ability to model the predictive confidence for OD and outlier samples. Among these methods, JL-P with knowledge distillation is shown to be the most effective regularization that can achieve low ECE and competitive raw quality at the same time. Nevertheless, it requires access to the corpus of the pre-training phase, which may not be available in some cases.

There is clearly more room for further improve these methods. Mixout exhibits a promising ability to model the confidence of OD and outlier samples properly and improve the OD generalization

---

[2]We present the results of JL-D without knowledge distillation and JL-P with knowledge distillation in Table 2. We show the full results for JL in A.3.1.

(e.g., the result on HellaSWAG) by taking advantage of the pre-trained models. However, Mixout fails to balance the preservation of pre-trained features and the learning of downstream tasks in some cases, which leads to low accuracy and high ECE. Figure 4 demonstrates that the outlier samples can be easily distinguished from the hidden representations of the JL-D models, but they do not provide as reasonable predictive confidence as JL-P, as shown in Figure 7. We leave these questions for future work.

# 6  RELATED WORK

**Uncertainty estimation of PLMs.** Previous works have demonstrated that PLMs can be beneficial for improving the robustness and calibration on downstream tasks compared to non-pre-trained models (Hendrycks et al., 2020; Desai & Durrett, 2020). However, PLMs can still fail to model their predictive uncertainty on downstream tasks. For example, Desai & Durrett (2020); Kong et al. (2020); Guo et al. (2021) have shown that fine-tuned masked language models (e.g., BERT (Devlin et al., 2019), RoBERTa (Liu et al., 2019)) are overconfident on text classification tasks, while Jiang et al. (2021) has shown that powerful generative PLMs (e.g., T5 (Raffel et al., 2020), BART (Lewis et al., 2020), and GPT-2 (Radford et al., 2019)) are poorly calibrated on QA tasks. In this work, we study a specific failure case of fine-tuned LM in which the models are overconfident of the OD and outlier samples due to catastrophic forgetting.

**Calibrating fine-tuned LMs.** Several approaches have been developed to calibrate the fine-tuned LMs on NLU tasks. For instance, Desai & Durrett (2020) demonstrate that temperature scaling and label smoothing can improve the calibration of the models in ID and OD settings, respectively. He et al. (2021b) introduce a new discriminative objective under the noise contrastive estimation (NCE) framework to jointly train an energy-based model defined on the classifier, which leads to better ID calibration. Fan et al. (2020); Zhang et al. (2021) model the attention weights as random variables and design a series of methods to optimize the stochastic attention layer with variational inference, which yields better performance in accuracy and calibration compared to vanilla deterministic attention layers. Kong et al. (2020); Park & Caragea (2022) adopt Mixup (Zhang et al., 2018) to calibrate fine-tuned language models and exhibit effectiveness in calibration under both ID and OD settings. We tackle the problem of calibrating the fine-tuned LMs from a new perspective by focusing specifically on the pre-training & fine-tuning paradigm and validate that preserving the pre-trained features is an effective way to improve the fine-tuned LMs' calibration.

**Benefit from mitigating catastrophic forgetting.** Previous works have shown that mitigating catastrophic forgetting of PLMs can be helpful for various aspects of downstream tasks. For example, Chen et al. (2020); Lee et al. (2020) show that constraining the models' parameters closer to the pre-trained ones can improve the training stability and performance of fine-tuned LMs on downstream tasks. Xie et al. (2021) validate that standard fine-tuning can destroy the output structure of pre-trained generative denoiser such as BART and show that preserving pre-trained features via lightweight fine-tuning can improve out-of-distribution generalization on downstream generation tasks. Dong et al. (2021) show that the pre-trained features of PLMs are beneficial for a robust objective model and improve the adversarial robustness of fine-tuned language models by maximizing the mutual information between the hidden representation of the pre-trained and fine-tuned models during the whole fine-tuning process. Our work specifically focus on uncertainty estimation of the fine-tuned LMs and makes a complementary contribution that the calibration of fine-tuned LMs can be improved by mitigating catastrophic forgetting.

# 7  CONCLUSIONS

In this work, we show that PLMs that pre-trained with large corpora are inherently well-calibrated on the MLM task while the fine-tuned LMs suffer from overconfidence due to catastrophic forgetting. Our experimental results validate that preserving the pre-trained features can better calibrate the fine-tuned LMs. We hope our work can draw more attention to the deeper exploitation of the pre-trained features learned by PLMs and contribute to building safe and reliable NLP systems for real-world applications.

REPRODUCIBILITY STATEMENT

We have provided detailed setup for all experiments in §3.1, §3.2, §5.1, A.1, and A.2, we submit our codes as the supplementary material. The information we provided is sufficient to reproduce our results.

ACKNOWLEDGMENTS

Thanks Peng Cui, Zhijie Deng, Wenbo Hu, Weize Chen, Xu Han for valuable discussions. This work was supported by the National Key Research and Development Program of China (2020AAA0106302); NSF of China Projects (Nos. 62061136001, 61620106010, 62076145, U19B2034, U1811461, U19A2081, 6197222); a grant from Tsinghua Institute for Guo Qiang; the High Performance Computing Center, Tsinghua University. J.Z was also supported by the XPlorer Prize.

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

# A   APPENDIX

## A.1   DATASET DETAILS

The general information of in-domain and out-of-domain datasets of the three NLU tasks are shown below:

**Natural Language Inference:**   Stanford Natural Language Inference (SNLI) (Bowman et al., 2015) requires the model to learn the textual entailment by predicting the relationship between given premise and hypothesis among *entailment*, *contradiction*, or *neutral*. Multi-Genre Natural Language Inference (MNLI) (Williams et al., 2018) shares the same task form as SNLI, but the samples are from more diverse domains than SNLI.

**Paraphrase Detection:**   Quora Question Pairs (QQP) (Shankar et al., 2017) contains question pairs from Quora. The model needs to discriminate whether the given pairs are semantically equivalent. TwitterPPDB (Lan et al., 2017) is the out-of-domain dataset that collects sentence pairs shared with the same URLs.

**Commonsense Reasoning:**   Situations With Adversarial Generations (SWAG) (Zellers et al., 2018) is a common sense reasoning task that requires the model to choose the most plausible continuation of a sentence given four possible candidates. HellaSWAG (Zellers et al., 2019) is designed as a more challenging commonsense reasoning task for the pre-trained language models built with Adversarial Filtering.

The statistic of the datasets for both MLM and NLU tasks are shown in Table 3. For the datasets of NLU tasks (SNLI/MNLI, QQP/TwitterPPDB, SWAG/HellaSWAG), we use the published version by Desai & Durrett (2020). For WikiText-103, we use the version provided by Huggingface Datasets (Lhoest et al., 2021) library. Note that the training splits of the OD datasets (MNLI, TwitterPPDB, HellaSWAG) are not used.

Table 3: The size of the training, validation, test splits and number of labels for all datasets.

| Dataset | #Train | #Validation | #Test | #Labels |
|---|---|---|---|---|
| SNLI (Bowman et al., 2015) | 549,368 | 4,922 | 4,923 | 3 |
| MNLI (Williams et al., 2018) | 392,702 | 4,908 | 4,907 | 3 |
| QQP (Shankar et al., 2017) | 363,871 | 20,216 | 20,217 | 2 |
| TwitterPPDB (Lan et al., 2017) | 46,667 | 5,060 | 5,060 | 2 |
| SWAG (Zellers et al., 2018) | 73,547 | 10,004 | 10,004 | 4 |
| HellaSWAG (Zellers et al., 2019) | 39,905 | 5,021 | 5,021 | 4 |
| WikiText-103 (Merity et al., 2016) | 1,801,350 | 3,760 | 4,358 | — |

## A.2   SETUPS

### A.2.1   SETUP FOR FULL FINE-TUNING AND PARAMETER-EFFICIENT TUNING

We conduct the experiments with the Huggingface Transformers library (Wolf et al., 2020). For parameter-efficient tuning methods, we use the implementations of OpenDelta (Ding et al., 2022) library for the three parameter-efficient tuning methods (Adapter, LoRA, Prefix Tuning). We use the same default hyperparameters provided by the OpenDelta library for each method across all three tasks [3].

All fine-tuning methods are trained with the AdamW optimizer (Loshchilov & Hutter, 2019). For full fine-tuning, we use a learning rate of 1e-5 across all tasks. For parameter-efficient tuning methods, we search the learning rate among {1e-5, 2e-5, 5e-5, 1e-4, 2e-4, 5e-4}. We use the one that has the best ID accuracy on the validation set, which is the widespread application scenario for the parameter-efficient tuning methods. Table 4 shows the learning rate used to fine-tune all the methods. For other training hyperparameters, we follow the default setup of the Huggingface Trans-

---

[3]The default hyperparameters can be found in the definition of each class on the document: `https://opendelta.readthedocs.io/en/latest/modules/deltas.html`.

formers library [4]. In specific, we set a batch size of 32, a maximum sequence length of 256, and a weight decay of 0.1. We use a linear decay learning rate scheduler without warmup and do not apply gradient clipping. Note that the reported baselines for full fine-tuning and Mixup (Desai & Durrett, 2020; Park & Caragea, 2022) in Table 2 do not use learning rate scheduler and use a gradient clip of 1.0 compared to our runs. There are also some minor differences, such as the padding strategy for input texts between our fine-tuning setup and theirs.

Table 4: Learning rate of fine-tuning methods on each task. The models are fine-tuned on the training split of SNLI for natural language inference (NLI), QQP for paraphrase detection (PD), SWAG for commonsense reasoning (CR).

| Task | Full-FT | Adapter | LoRA | Prefix Tuning |
|------|---------|---------|------|---------------|
| NLI | 1e-5 | 2e-4 | 2e-4 | 1e-4 |
| PD | 1e-5 | 2e-4 | 2e-4 | 1e-4 |
| CR | 1e-5 | 1e-4 | 1e-4 | 2e-4 |

### A.2.2 SETUP FOR THE JOINT LEARNING METHOD

We conduct hyperparameter search with the ID/OD validation set for the models that joint learning with the MLM objective (JL). Specifically, we set a learning rate of 1e-5 and a batch size of 32 across all three tasks as Desai & Durrett (2020) does, except for fine-tuning JL-P with label smoothing on SWAG, where we use a larger learning rate of 5e-5. For other training parameters, we adopt the same setup described in A.2.1 except for fine-tuning JL-P on QQP, where we do not use a learning rate scheduler. For the hyperparameter of the JL models, we search the scaling factor of the MLM loss $\alpha_{\mathrm{mlm}} \in \{0.1, 0.3, 0.5, 1, 2, 3, 4, 5\}$, the coefficient of the regularization term on contextualized representation $\beta_{L^2} \in \{$1e-9, 1e-8, 1e-7, 1e-6, 1e-5, 1e-4$\}$ , the masking probability for a sentence $p_{\mathrm{mask}} \in \{0.05, 0.15, 0.3, 0.4, 0.5, 0.6\}$, and the hyperparameter of label smoothing $\sigma_{\mathrm{ls}} \in \{0.01, 0.03, 0.05\}$ for each task. We also search the maximum sequence length of the MLM task for JL-P and the batch size of MLM tasks for both JL-D and JL-P. In detail, we use a batch size for the MLM task of 32 on NLI and PD tasks and a batch size for the MLM task of 8 on the CR task. The maximum sequence lengths of the MLM task for JL-P are set to 32/32/64 for NLI, PD, and CR tasks, respectively. Training a single model for 3 epochs can be done in two hours with a single NVIDIA A40 48G GPU. We present the detailed hyperparameter setup of each method on each task in Table 5.

Table 5: Hyperparameters of JL models for each NLU task.

| | $\alpha_{\mathrm{mlm}}$ | $p_{\mathrm{mask}}$ | $\beta_{L^2}$ | $\sigma_{\mathrm{ls}}$ |
|------|------|------|------|------|
| **Task: SNLI/MNLI** | | | | |
| JL-D | 0.3 | 0.4 | 1e-5 | — |
| JL-P | 0.3 | 0.4 | 1e-5 | — |
| JL-P + LS | 0.5 | 0.6 | 1e-8 | 0.03 |
| **Task: QQP/TwitterPPDB** | | | | |
| JL-D | 1 | 0.15 | 1e-5 | — |
| JL-P | 4 | 0.15 | 1e-7 | — |
| JL-P + LS | 4 | 0.15 | 1e-9 | 0.01 |
| **Task: SWAG/HellaSWAG** | | | | |
| JL-D | 1 | 0.3 | 1e-9 | — |
| JL-P | 3 | 0.3 | 1e-9 | — |
| JL-P + LS | 3 | 0.05 | 1e-4 | 0.05 |

### A.2.3 SETUP FOR REGULARIZATION WITH PRE-TRAINED WEIGHT

**Pre-trained Weight Decay:** We tune the regularization strength $\lambda_{\mathrm{PWD}} \in \{0.1, 1, 10, 20, 50, 100\}$ and use $\lambda_{\mathrm{PWD}} = 10/20/1$ for SNLI, QQP, and SWAG, respectively.

---

[4]The NLI and PD tasks correspond to the text classification setting, while the CR task corresponds to the multiple choice criterion of huggingface transformers library.

**Mixout:** We tune the mixout probability $p_{\mathrm{mixout}} \in \{0.1, 0.3, 0.5, 0.7, 0.9\}$ and use $p_{\mathrm{mixout}} = 0.9$ for all three NLU task.

For other training parameters, we use the same default setup described in A.2.2.

### A.2.4    COMPARISON OF TRAINING TIME

We compare the training time of different methods using a single NVIDIA A40 48G GPU. We use the full fine-tuning as a baseline (1x). The time cost of 3 training epochs using full fine-tuning is 1/0.8/0.3 GPU hours for SNLI/QQP/SWAG. We present the time cost of different methods in Table 6.

Table 6: Comparison of training time of different methods.

| Task | Full-FT | Parameter-Efficient Tuning | PWD | Mixout | JL |
|---|---|---|---|---|---|
| NLI & PD | 1x | 0.6x∼1x | 1x | 2.4x | 2x |
| CR | 1x | 0.75x | 1x | 1.5x | 1.5x |

## A.3 ADDITIONAL EXPERIMENTAL RESULTS

### A.3.1 ABLATION STUDY FOR THE JOINT LEARNING METHOD

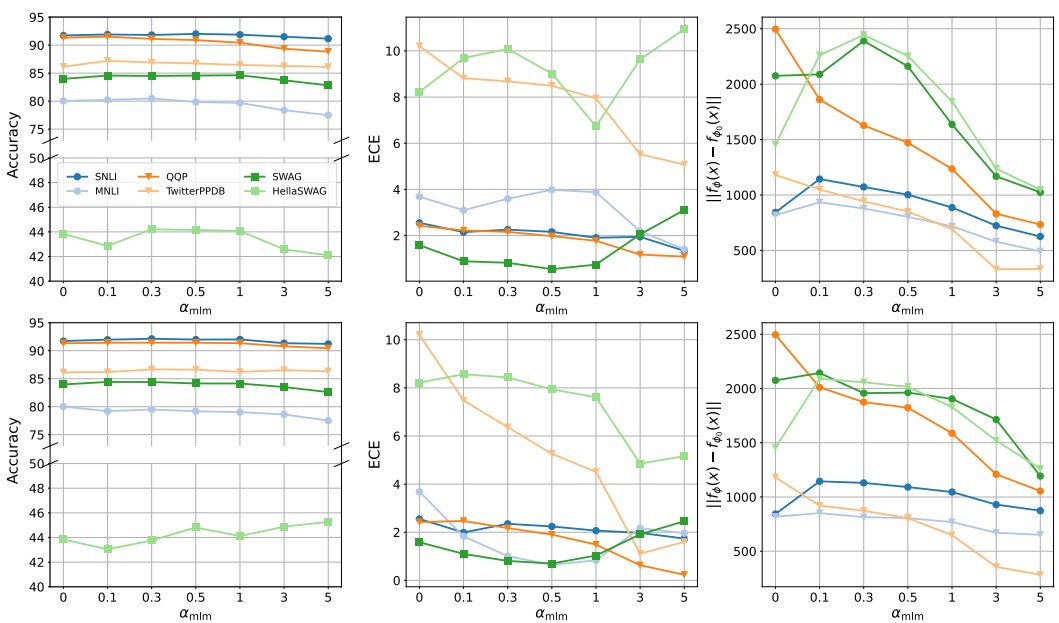

Figure 6: Accuracy, ECE and $L^2$ distance to the pre-trained hidden representations ($||f_\phi(\boldsymbol{x}) - f_{\phi_0}(\boldsymbol{x})||$) of JL-D (up) and JL-P (down) over different scaling factors $\alpha_{\mathrm{mlm}}$ of the MLM Loss in ID and OD settings.

**Effect of the MLM objective:** As shown in Figure 6, compared to vanilla fine-tuning ($\alpha_{\mathrm{mlm}} = 0$), introducing the MLM objective lowers the ECE effectively in both ID and OD settings with a relatively small effect on accuracy. As the magnitude of the MLM loss increases, the features of the fine-tuned models are closer to those of the pre-trained models, reflected in both the geometry of feature space (shown in Figure 4) and euclidean distance between the representations of pre-trained and fine-tuned models (shown in Figure 6), and the ECE of the fine-tuned models decreases. However, when the weight of the MLM loss becomes too large compared to the classification loss, the performance of the NLU task will be apparently damaged. For the overconfidence toward the OD samples and outlier samples illustrated in §3.2, as shown in Figure 7, can be mitigated by using a relatively small magnitude of $\alpha_{\mathrm{mlm}}$, while increasing it can have a more significant effect.

**Effect of introducing corpus of pre-training phase:** From Table 2 and Figure 6, we observe that performing the MLM task with the corpus of the pre-training phase has lower OD ECE on all three tasks. This confirms our belief that using the corpus of the pre-training phase can preserve more helpful features from the PLMs. We also notice that sometimes JL-P degrades the ID calibration (e.g., calibration on SWAG in Table 2), however, applying label smoothing on downstream tasks can relieve this negative effect. An interesting observation is that although the hidden representations given by both JL-D and JL-P can distinguish outlier samples easily, the JL-D holds higher confidence toward outlier samples than JL-P. As shown in Figure 7, compared to JL-P, the JL-D are more confident in samples from the BookCorpus (Zhu et al., 2015) dataset that is not seen by both JL-P and JL-D in the fine-tuning phase for a fair comparison, which suggests utilizing the pre-training phase's corpora that are more diverse than the training datasets are helpful for the fine-tuned LMs to model the confidence of the outlier samples better.

**Effect of knowledge distillation from the pre-trained model:** Table 7 shows that applying knowledge distillation has limited effect on JL-D but enhances the accuracy and ECE of JL-P in most cases. We hypothesize that performing the MLM task with the corpus that is distinct from the downstream datasets can hurt relatively more performance on the downstream tasks, and distilling from the pre-trained model's predictive distribution might be a smoother and more effective regularization.

**Effect of regularization on the contextualized representation:** As shown in Figure 8, the introduced heuristic regularization term could improve the ECE of JL-D models under both ID and OD settings by smoothing the models' predictive confidence. The effect of this regularization term is similar to applying temperature scaling under the ID validation set, where large magnitudes can lead to overly conservative prediction confidence. We also find that a proper choice of $\beta_{L^2}$ can marginally benefit the accuracy under ID and OD settings. However, the effect of this term on Full-FT models is limited.

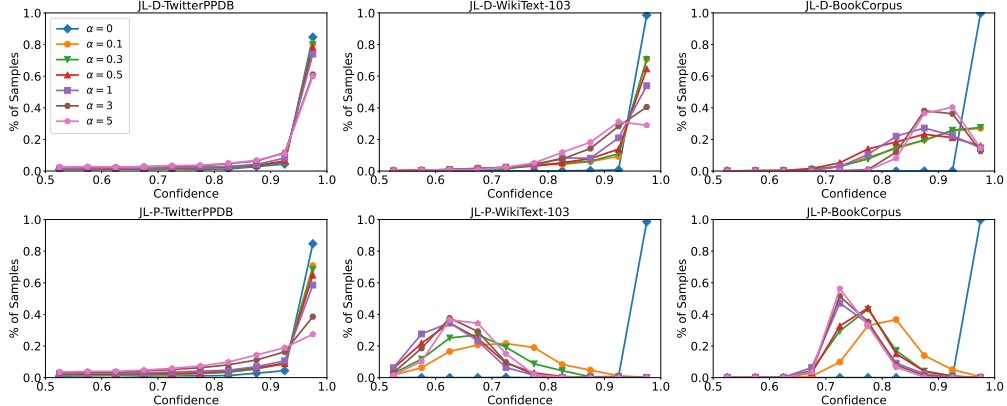

Figure 7: Confidence histogram for OD (TwitterPPDB) samples and two sets of outlier (WikiText-103, BookCorpus) samples of JL-D and JL-P.

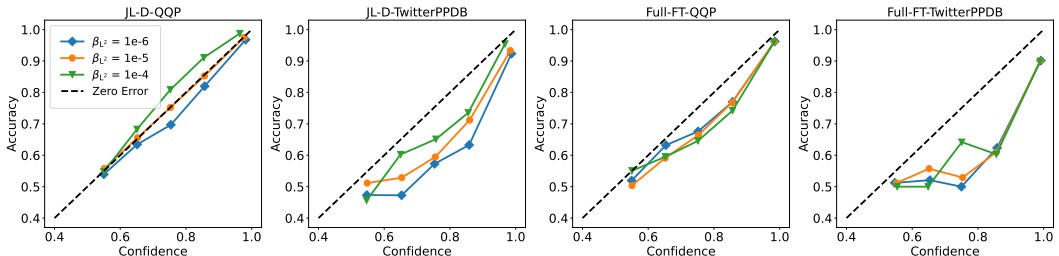

Figure 8: Reliability diagram of JL-D and Full-FT models with different $\beta_{L^2}$ values.

Table 7: The results with/without knowledge distillation for JL models.

| RoBERTa-base | In-Domain | | Out-of-Domain | |
|---|---|---|---|---|
| | Acc | ECE | Acc | ECE |
| **Task: SNLI/MNLI** | | | | |
| JL-D (w/o KD) | $91.95_{0.12}$ | $0.67_{0.05}$ | $79.85_{0.29}$ | $1.26_{0.50}$ |
| JL-D (w/ KD) | $91.64_{0.19}$ | $0.75_{0.21}$ | $79.99_{0.30}$ | $1.31_{0.28}$ |
| JL-P (w/o KD) | $91.72_{0.14}$ | $1.12_{0.17}$ | $79.45_{0.25}$ | $1.19_{0.17}$ |
| JL-P (w/ KD) | $91.74_{0.15}$ | $1.09_{0.15}$ | $79.36_{0.27}$ | $1.20_{0.43}$ |
| **Task: QQP/TwitterPPDB** | | | | |
| JL-D (w/o KD) | $90.42_{0.08}$ | $0.57_{0.12}$ | $86.67_{0.22}$ | $6.49_{0.44}$ |
| JL-D (w/ KD) | $90.86_{0.04}$ | $0.50_{0.08}$ | $86.26_{0.35}$ | $6.06_{0.52}$ |
| JL-P (w/o KD) | $89.75_{0.19}$ | $0.78_{0.47}$ | $86.77_{0.61}$ | $2.86_{1.01}$ |
| JL-P (w/ KD) | $90.50_{0.09}$ | $0.91_{0.50}$ | $86.50_{0.90}$ | $1.27_{0.39}$ |
| **Task: SWAG/HellaSWAG** | | | | |
| JL-D (w/o KD) | $84.49_{0.24}$ | $0.91_{0.25}$ | $43.75_{0.92}$ | $7.64_{0.43}$ |
| JL-D (w/ KD) | $84.03_{0.14}$ | $0.58_{0.08}$ | $44.68_{0.49}$ | $10.08_{0.36}$ |
| JL-P (w/o KD) | $82.39_{0.03}$ | $2.85_{0.11}$ | $42.68_{0.24}$ | $7.64_{0.37}$ |
| JL-P (w/ KD) | $83.51_{0.05}$ | $1.94_{0.12}$ | $44.78_{0.22}$ | $5.23_{0.43}$ |

### A.3.2 SAMPLING FROM JOINT LEARNING MODELS

Since the JL model is encouraged to learn both discriminative and generative representations, we can perform conditional generation with the MLM head $g_\phi$ and classifier $h_\varphi$. In this work, we choose Mask-Predict (Ghazvininejad et al., 2019) as our basic sampling algorithm, which can incorporate the rejection sampling framework to generate samples from the conditional distribution $p(\boldsymbol{x}|y^*)$ given the desired label $y^*$. We show the detailed sampling process in Alg 1.

---

**Algorithm 1:** Mask-Predict with Rejection Sampling

1 **Input:** Number of iterations $T$, Text Encoder $f_\phi$, MLM head $g_\theta$, Classifier $h_\varphi$, Desired label $y^*$, Number of initial masked tokens $N$.

2 **Initialize:** $\boldsymbol{x}^{(0)} \leftarrow [\texttt{MASK}]^N$

3 **for** $t \leftarrow 0$ **to** $T - 1$ **do**                                   // Mask-Predict iteration

4     $n \leftarrow f(N, t)$                        // Determine number of tokens to mask at iteration $t$

5     **repeat**                                              // Rejection sampling loop

6        $\hat{\boldsymbol{x}}^{(t)} \leftarrow g_\theta(f_\phi(\boldsymbol{x}^{(t)}))$      // Select prediction with highest probability for every $[\texttt{MASK}]$ token.

7        $\mathbb{M}_t \leftarrow \arg\min_i(p_{\text{mlm}}(x_i = \hat{x}_i^{(t)} | \boldsymbol{x}^{(t)}), n)$

8        $\hat{\boldsymbol{x}}_{\mathbb{M}_t}^{(t)} \leftarrow [\texttt{MASK}]$                  // Mask $n$ tokens with the lowest probability scores

9     **until** $u \sim \mathcal{U}(0, 1)$, $u \leq \text{Softmax}(h_\varphi(\hat{\boldsymbol{x}}^{(t)}))[y^*]/\tau$

10    $\boldsymbol{x}^{(t+1)} \leftarrow \hat{\boldsymbol{x}}^{(t)}$

11 **end**

12 **Output:** $\boldsymbol{x}^{(T)} \sim p(\boldsymbol{x}|y^*)$

---

Table 8 shows the samples generated by JL-D models with the sampling algorithm described above. The JL-D models are able to generate text with the given labels, which can be seen as a diagnostic for the model. For example, the models prefer to generate contradictory hypotheses by changing the objective's entity or adjectives with high confidence and tend to copy the prefix to generate positive textual entailment, which may expose some spurious correlations that the models rely on when performing the NLU tasks (Tu et al., 2020).

Table 8: Samples generated by JL-D models with SNLI and QQP test set using the Mask-Predict algorithm with rejection sampling. For SNLI, the model generates hypotheses given class labels {none (-), entailment, contradiction, natural} and premise prefixes. For QQP, the model generates sentences given class labels {none (-), non-paraphrase (non-para), paraphrase (para)} and question prefixes. We mark the generated samples whose assigned label is consistent with the given label using [ ] and mark the failure cases where the assigned label is not consistent with the given label using [ ]. We also report the corresponding confidence of the class of the generated text after the label.

| | |
|---|---|
| **Text Prefix:** | *A mountain biker rides up a hill on a red bicycle.* |
| **[ - ]** | *A mountain biker rides a bike on a hill.* |
| **[ Entailment, 99% ]** | *A mountain biker rides a bike up a hill.* |
| **[ Contradiction, 99% ]** | *A mountain biker rides **downhill** on a **blue** bicycle.* |
| **[ Natural, 65% ]** | *A mountain biker is trying to climb a hill.* |
| **Text Prefix:** | *A man plays the french horn as his pianist plays the supporting melody on stage.* |
| **[ - ]** | *A man is playing a french horn for a concert.* |
| **[ Entailment, 97% ]** | *A man is playing a french horn on a stage.* |
| **[ Contradiction, 99% ]** | *A man is playing a **flute** for a **crowd**.* |
| **[ Natural, 91% ]** | *A man is playing a song on a concert stage.* |
| **Text Prefix:** | *Two young men in unusual clothing are jumping in a gym.* |
| **[ - ]** | *Two men are playing basketball.* |
| **[ Entailment, 96% ]** | *Two men are jumping around.* |
| **[ Contradiction, 90% ]** | *Two men are jumping **outside**.* |
| **[ Natural, 44% ]** | *Two men are playing basketball.* |
| **Text Prefix:** | *a blue and gray race car driving on a dirt track.* |
| **[ - ]** | *A race car is driving on a dirt track.* |
| **[ Entailment, 98% ]** | *A race car is driving on a dirt track.* |
| **[ Contradiction, 99% ]** | *A race car is **parked** on a dirt track.* |
| **[ Natural, 66% ]** | *A race car is racing on a dirt track.* |
| **Text Prefix:** | *A boy poses in karate form and uniform.* |
| **[ - ]** | *A boy is practicing karate.* |
| **[ Entailment, 75% ]** | *A boy is practicing karate.* |
| **[ Contradiction, 2% ]** | *A boy is in a costume.* |
| **[ Natural, 25% ]** | *A boy is practicing karate.* |
| **Text Prefix:** | *Four females wearing helments are riding on an ATV.* |
| **[ - ]** | *Four females are wearing helments on an ATV.* |
| **[ Entailment, 98% ]** | *Four females are wearing helments on an ATV.* |
| **[ Contradiction, 99% ]** | *Four females are **riding on a horse** in the park.* |
| **[ Natural, 99% ]** | *Four females are riding a ATV in the desert.* |
| **Text Prefix:** | *How does cloud computing work?* |
| **[ - ]** | *How does cloud computing in India work?* |
| **[ Non-Paraphrase, 95% ]** | *How does Google think cloud computing work?* |
| **[ Paraphrase, 74% ]** | *How does I understand cloud computing work?* |
| **Text Prefix:** | *Do you think time travel is possible?* |
| **[ - ]** | *Do you think space and time travel is possible?* |
| **[ Non-Paraphrase, 73% ]** | *Do you think gravity can make time travel possible?* |
| **[ Paraphrase, 98% ]** | *Do you think it is possible to time travel?* |
| **Text Prefix:** | *What is a good data analysis book?* |
| **[ - ]** | *What is a good free data analysis book?* |
| **[ Non-Paraphrase, 78% ]** | *What is a good book for analysis books?* |
| **[ Paraphrase, 67% ]** | *What is the best free data analysis book?* |
| **Text Prefix:** | *Feeling bored. What do I do?* |
| **[ - ]** | *What to do to get bored?* |
| **[ Non-Paraphrase, 97% ]** | *What should I do to myself?* |
| **[ Paraphrase, 28% ]** | *What should I start to do?* |

