# OpenReview forum: "Preserving Pre-trained Features Helps Calibrate Fine-tuned Language Models"
_ICLR.cc/2023/Conference — ICLR 2023 poster_

### Official Review · Reviewer_eTyr · 2022-10-25

**Confidence:** 3
**Correctness:** 3
**Technical Novelty And Significance:** 2
**Empirical Novelty And Significance:** 3
**Recommendation:** 6

**Clarity, Quality, Novelty And Reproducibility:**

As mentioned above, the presentation of the key technical contribution could be vastly simplified by forgoing the DVAE formalism. Otherwise, the paper is easy to understand and generally well-written. The novelty is somewhat limited given that the core idea is simply including the MLM objective during finetuning. This can be strengthened by considering other ways to preserve pretraining features, such as “weight decay” back to the pretrained weights and prior work on combatting catastrophic forgetting. If including the MLM objective is indeed to best way to preserve pretraining features for better calibration, the conclusion of this paper would be much informative.

**Strength And Weaknesses:**

Model calibration is often overlooked in the context of large language models. This paper makes the intriguing observation that efficient finetuning methods yield better calibrated models than full finetuning, likely due to the former’s ability to preserve pretraining features.

This work leverages this observation to further improve calibration by retaining the pretraining objective during finetuning. However, the denoising variational auto-encoding interpretation seems unnecessary. The referenced paper, Im et al., 2017, is motivated by the ability to obtain a more expressive variational distribution by marginalizing over input corruptions (e.g., to obtain a mixture of Gaussians). This marginalization makes the usual ELBO intractable, which motivated their new DVAE training criterion.

This work, however, uses a deterministic mapping from x to z, i.e., the variational distribution is the Dirac Delta distribution. By forgoing the stochasticity in the latent space, this work neither shares the core motivation of nor benefits from the DVAE paper. It is perhaps okay to mention Denoising Autoencoder in this context, but it basically describes the MLM objective which is not a contribution of this paper.

The method proposed in this work is better understood by inspecting Eq. 6 and can be summarized as “performing MLM on the finetuning data in addition to the finetuning objective (plus an L2 regularization in the latent space).” It is also unclear what the role of the L2 regularization is since it is present in neither the usual pretraining nor the usual finetuning objective. Indeed, as shown in Table 8, successful values of beta are many orders of magnitude smaller than those of alpha.

I would encourage the authors to reframe their method without the DVAE framework and draw from the established literature on catastrophic forgetting. Moreover, given the baseline success of efficient finetuning methods, it seems worthwhile to investigate how they can be made better calibrated for scenarios where calibration is much more important than a slight performance drop.


**Summary Of The Paper:**

This work observes that while a pretrained language model is well-calibrated on the pretraining task as measured by ECE, it becomes poorly calibrated on downstream tasks after finetuning. It is further observed that efficient finetuning methods, such as Adapter, LoRA, and Prefix Tuning, result in better calibrated models than full finetuning. This leads the authors to conclude that the source of the miscalibration is the distortion of features learned during pretraining.

The authors then proposed to include the pretraining objective during finetuning to preserve pretraining features. This improves model calibration as measured by ECE while largely retaining performance on SNLI/MNLI, QQP/TwitterPPDB, and SWAG/HellaSWAG.

**Summary Of The Review:**

Great motivation and interesting observations. However, the presentation can be simplified by forgoing the DVAE formalism, which adds little to this work. This paper can be further strengthened by considering other baselines that preserve pretraining features.

---

> ### Author Response · Authors · 2022-11-16
> **Response to Reviewer eTyr**
>
> Thank you for the supportive comments and constructive feedback! We address some comments below:
>
> Q1: The DVAE interpretation seems unnecessary.
>
> A1: We agree that the DVAE framework contributes little to the key idea that preserving the pre-trained features can help the calibration of the fine-tuned LMs. We decide to reframe the paper without the DVAE framework.
>
> Q2: Adding more baseline methods that preserves pretraining features.
>
> A2: Thank you for the constructive advice! We move the parameter-efficient tuning to the methods section as a baseline. And we add two new baselines: Weight decay toward pre-trained weight and Mixout [1]. The experimental results illustrate that all the baseline methods, including parameter-efficient tuning, pre-training weight decay, Mixout, and joint learning with MLM loss (the old DVAEs), can improve the ECE of the fine-tuned LMs. The results further support our hypothesis that preserving pre-trained features helps calibrate the fine-tuned language models. In our revised version, we also compare how these methods affect the model's predictive confidence differently.
>
> Q3: It is unclear what the role of the L2 regularization in the latent space is.
>
> A3: Previous work [2] and Figure 2 in our revised version show that fine-tuning tends to enlarge the distance between the hidden representations of samples. We treat it as a heuristic regularization term that prevents the fine-tuned model's hidden representations become too distant from each other in our revised version. In practice, we found that this term can smooth the predictive confidence of the model with the joint learning method and slightly improve classification performance in some cases. We also add the ablation study of this term to the appendix.
>
> Q4: It seems worthwhile to investigate how efficient finetuning methods can be made better calibrated for scenarios where calibration is much more important than a slight performance drop.
>
> A4: The calibration of parameter-efficient tuning methods is indeed an interesting topic. In our experiments, we found that using a more conservative learning strategy (e.g., lower learning rates) leads to even better calibration results than presented in the paper. However, the drop in performance is also more significant. We also observe a similar phenomenon in Mixout, where applying a high Mixout probability yields more reasonable predictive confidence toward the OD and outlier samples, but the performance is damaged and unstable. It is a promising future research direction to achieve better predictive confidence and maintain competitive performance for these methods.
>
> [1] [Mixout: Effective Regularization to Finetune Large-scale Pretrained Language Models](https://arxiv.org/abs/1909.11299) (Lee, Cheolhyoung, Kyunghyun Cho, and Wanmo Kang, ICLR 2020)
>
> [2] [A Closer Look at How Fine-tuning Changes BERT](https://aclanthology.org/2022.acl-long.75) (Zhou & Srikumar, ACL 2022)

---

> > ### Comment · Reviewer_eTyr · 2022-11-22
> > **Response**
> >
> > Very nice.
> >
> > The authors did a good job improving the manuscript in a short period of time. The conclusion is now easier to understand and better supported.
> >
> > I think this paper should be considered for acceptance as a result.

---

> > > ### Author Response · Authors · 2022-11-22
> > > **Thanks!**
> > >
> > > Thank you again for your efforts and acknowledgment of our work. The improvements can't be made without your valuable reviews and constructive suggestions.

---

### Official Review · Reviewer_YFTC · 2022-10-25

**Confidence:** 4
**Clarity, Quality, Novelty And Reproducibility:** 1, The paper in general is clearly wr…
**Correctness:** 4
**Technical Novelty And Significance:** 3
**Empirical Novelty And Significance:** 3
**Recommendation:** 6

**Strength And Weaknesses:**

Strength: the paper studies ECE in language models which is somewhat of an overlooked area.

Weaknesses:
1, The paper did not provide sufficient motivation for why the broader community should care about a metric like ECE. They have a few references and some vague description of "disastrous consequences in safety critical applications". But they did not concretely describe such applications and such consequences or demonstrate that this is the most important issue of language models when applied in such applications. Thus this reviewer is not convinced that they are solving a problem of high enough priority.

2, The paper did not go deep into the reason for why finetuning causes over-confidence in language models. They observed this phenomenon and proposed the most basic solution, which is mixing finetuning with pretraining. The result is also non-surprising. When the finetuning is mixed with the pretraining, then the resulting model behaves a bit more similar to the pretrained model. One might imagine that the overconfidence and the regression of the finetuned model as demonstrated in Table 1 is a result of the lack of diversity in the finetuning dataset. It would be interesting to establish the connection between the diversity of the training dataset and the ECE/overconfidence issue empirically, which would be a meaningful contribution to the community in understanding the response of deep networks to different data distributions.

3, The discussion about parameter efficient finetunings are interesting. However if one wants to make these model still do MLM, one can just remove the trained small parameter components and keep the pretrained model. The motivation for testing a parameter efficient model that is finetuned on specific downstream task on MLM is not very clear.

**Summary Of The Paper:**

The paper studies the ECE of language models trained using different recipes, including the pretrained model, finetuned model and parameter-efficient finetuned models. They find that the finetuned model has larger ECE and tend to be over confident when making predictions. The paper proposes a baseline method of mixing the finetuning objective and the pretraining objective during finetuning to address this and this method works to lower ECE.

**Summary Of The Review:**

The paper has made some good observations on how LMs training differently would have different ECE and overconfidence behavior. They proporsed a baseline mixture of pretrain and finetune to reduce ECE for finetuning, which was successful. It was not clear why this is an important discovery or innovation that significantly benefits the ICLR community.

---

> ### Author Response · Authors · 2022-11-16
> **Response to Reviewer YFTC (Part 1/2)**
>
> Thank you for the valuable review and constructive comments! We address several comments below.
>
> Q1: The paper did not provide sufficient motivation for why the broader community should care about a metric like ECE.
>
> A1: Thanks for pointing out the deficiency in our presentation of the core motivation. The PLMs fine-tuned on NLU tasks have been experimented with for decision-making in real-world applications. For example, [1] uses pre-trained BERT to conduct the medical language inference task, where the fine-tuned LM predicts the type of disease given the description of symptoms. As a decision maker in a real-world system, the LM-based classifier need not only be accurate but also should provide calibrated confidence along with the prediction. Back to the medical inference example, if the LM is well-calibrated, i.e., it knows what it does not know, then the wrong decisions made by LM are likely to correspond with low confidence, which can be easier corrected by human experts and is helpful to build trustiness between the ML system and human beings. However, the fine-tuned language models are shown to suffer from overconfidence, especially under out-of-domain settings, which creates concerns for their deployment in real-world applications. Hence, investigating why the fine-tuned LMs are miscalibrated and how to improve their calibration is a worthwhile problem for building safe and reliable NLP systems for real-world applications. We will address the core motivation more at the beginning of the updated version.
>
> Q2: The paper did not go deep into the reason for why finetuning causes over-confidence in language models.
>
> A2: Thanks for the critical comments, which are helpful for us to improve our work! We provide more empirical results and intuitive explanations to illustrate why fine-tuning causes over-confidence in the revised version. In short, we first show that in addition to being well-calibrated, the PLMs' ECE is relatively stable under larger level of corruption on the inputs. Intuitively, this suggests that the pre-trained features can provide robust predictive confidence under domain shift, which might be helpful to the calibration of the downstream classification task. However, the fine-tuned language models do not take advantage of the pre-trained features due to catastrophic forgetting. They forget the informative pre-trained features that can easily distinguish different domains and project the ID, OD, and undefined outlier samples to a much simpler data manifold than the PLMs, which makes them exhibit unusual overconfidence toward the OD and outlier samples. For example, we show that the LM fine-tuned on the QQP task for three epochs makes predictions with nearly 100% confidence given a dataset with no relationship to the learned downstream classification task. Our revised version draws a complete picture of these through empirical results.
>
>
> [1] [Infusing Disease Knowledge into BERT for Health Question Answering, Medical Inference and Disease Name Recognition](https://aclanthology.org/2020.emnlp-main.372) (He et al., EMNLP 2020)

---

> > ### Author Response · Authors · 2022-11-16
> > **Response to Reviewer YFTC (Part 2/2)**
> >
> > Q3: The proposed solution is basic and the result is non-surprising. It was not clear why this is an important discovery or innovation that significantly benefits the ICLR community.
> >
> > A3:
> > We would like to address that our main contribution is neither the simple & basic solution itself nor the fact that the proposed methods are more close to the pre-trained model.
> > Indeed, our proposed method is simple, and making it more similar to the pre-trained model is our most basic expectation of it. There may be some overemphasis on this obvious point in our presentation, and we will pay more attention to the proposed methods' impact on predictive confidence and calibration in the revised version.
> >
> > However, making the fine-tuned model closer to the pre-trained model (i.e.,  preserving the pre-trained features) can effectively improve the calibration on downstream classification tasks is interesting and surprising. The fact that **the PLMs are well-calibrated on the MLM task** does not directly affect **the calibration of the fine-tuned model on downstream tasks**.
> >
> > In this work, we first explore how the disparity between the pre-trained and fine-tuned model impacts the calibration of the fine-tuned LMs on downstream tasks (as addressed in A2) and validate that preserving the pre-trained features is an effective and competitive regularization method for improving the calibration on downstream task compared to existing calibration methods. We argue that these findings open a new avenue for improving the calibration of fine-tuned models by leveraging the PLM and, subsequently, the massive amount of pre-trained data. The positive results of these simple methods further strengthen the potential of our new perspective, and the result might be improved by more sophisticated future research.
> >
> > Besides, to further enrich the method part, we add two new baselines that preserve pre-trained features, including weight decay toward the pre-trained weight and Mixout [2]. We show that all adopted methods can improve the calibration of the fine-tuned LMs on downstream tasks, which further validates our hypothesis.
> >
> > Q4: The motivation of testing a parameter efficient model on MLM is not very clear.
> >
> > A4: To answer this question, we need to go back to why we use the parameter efficient model first. We aim to find the fine-tuned models that preserve the pre-trained features and examine whether they can improve calibration on the downstream task. A natural choice is the parameter-efficient models since they keep most pre-trained weights. So in our experiments, the parameter efficient models are treated as a whole fine-tuned model, and we validate whether they preserve the pre-trained features through the MLM task.
> >
> > However, as you have commented, the observation that the calibration of parameter-efficient tuning is better than the full fine-tuning is just a phenomenon. In our revised version, we focus on answering how the disparity between pre-trained and fine-tuned model affect calibration on the downstream task in this section. We move the discussion of parameter-efficient tuning to the method section along with other methods that can preserve the pre-trained features.
> >
> > Q5: Relationship between data diversity of the training dataset and overconfidence.
> >
> > A5: Thank you for providing a good point of view on why fine-tuned LM exhibits overconfidence. We agree that data diversity is an important factor in the fine-tuned model's calibration. Figure 2 in our modified version shows that the fine-tuned model's hidden representation lies on a simpler data manifold than the pre-trained model, which might be caused by the training set's lack of data diversity.  In this work, we aim to mitigate the miscalibration due to the lack of data diversity in the fine-tuning phase by leveraging the pre-trained features that are learned with enormous unsupervised corpora.
> >
> > Although no direct experiments are conducted on the diversity of the training data set, our experimental results show that performing the MLM task with the pre-training phase's corpora while fine-tuning can better calibrate the fine-tuned LM under out-of-domain settings. In our revised version, we show that despite both JL-D and JL-P (the new name for the DVAEs) can distinguish the outlier samples from the hidden representations, the model trained with large corpora (JL-P) can provide better confidence to the OD and outlier samples, which might be a benefit of the diversity of the pre-training phase's corpora.
> >
> > [2] [Mixout: Effective Regularization to Finetune Large-scale Pretrained Language Models](https://arxiv.org/abs/1909.11299) (Lee, Cheolhyoung, Kyunghyun Cho, and Wanmo Kang, ICLR 2020)

---

> > > ### Comment · Reviewer_YFTC · 2022-11-21
> > > **The author has adequately addressed the question about motivation**
> > >
> > > After reading the response carefully, I believe the authors has addressed my biggest concerns in terms of motivation, both in the response and in the revised version of the paper. As such I believe that with this motivation established, the authors provided a thorough study of the phenomenon. Their observations associated with this problem is a useful contribution to the ICLR community.

---

> > > > ### Author Response · Authors · 2022-11-22
> > > > **Thanks!**
> > > >
> > > > We are glad to see that your concerns have been addressed after the rebuttal. Thank you for the careful re-evaluation and for recognizing the contribution of our work.

---

### Official Review · Reviewer_Kwo1 · 2022-10-28

**Confidence:** 3
**Correctness:** 3
**Technical Novelty And Significance:** 2
**Empirical Novelty And Significance:** 2
**Recommendation:** 6

**Clarity, Quality, Novelty And Reproducibility:**

### Quality
- The experiments are clear and well done.

### Clarity
- The paper itself has one (seemly important) section whose connection to the rest of the paper is unclear.

### Originality
- The idea of training classification together with MLM is not novel.
- Studying the impact of this on the calibration error is novel, but the scope/impact of this is quite narrow.


**Strength And Weaknesses:**

## Strengths
- The task of calibrating the results of a large language model is important and necessary in some practical applications.
- The observation that the problem of mis-calibration is created during fine-tuning is interesting and novel.
- The experimentation was done on multiple datasets and in two different settings. The authors run fine-tuning 5 time and report average and standard deviation, showing a more complete picture compared to the results of a single run.

## Weaknesses
- It is unclear why post-processing techniques, such as temperature scaling, are insufficient, that we need a new training paradigm for overcoming this post processing step, especially that temperature scaling is quite lightweight. This makes the overall contribution quite limited.
- Using MLM loss in addition to the standard classification loss is known (for a long time) to produce better results. Most work run an additional pre-training step on the target dataset, e.g., https://arxiv.org/pdf/1905.05583.pdf, https://arxiv.org/pdf/2004.11493.pdf, but joint training is also a known technique (which performs slightly worst, so less cited. Still, it was already published on e.g., https://dl.acm.org/doi/pdf/10.1145/3437963.3441777). This makes the comparison to the baseline unfair, since additional (unsupervised) training data was used.
- It is unclear what is the performance implications are on the training time. If the training scheme alternates between MLM and classification losses, this would result in 2x longer training time, which is a significant impact. This should be at least clarified in the paper.
- It is unclear how Section 5.1 is related to the rest of the paper (except for naming the algorithm), given that after all complex formulas the paper uses the standard MLM loss, and then move to knowledge distillation loss. Clarifying the connection to the rest of the paper (or dropping this part) would make it easier to follow the paper's contribution.
- Table 1, showing that after fine-tune the model is unable to do the original task, is not by itself surprising or interesting. The question is how this is impacting calibration on the classification task, which is not directly related to the quality of the model on the MLM task.

**Summary Of The Paper:**

The paper studies the calibration problem of BERT-like models on classification tasks. They show that the problem is created mainly during fine-tuning - the pre-trained model itself is well calibrated on the MLM task it is trained on. This MLM knowledge and calibration is destroyed during fine-tuning (a process called catastrophic forgetting in the literature).

To solve the calibration problem, the authors propose to continue fine-tune the model on the MLM task jointly with the actual classification
task at hand.

Experimentations on 3 datasets and both in-domain and out-of-domain settings show that the proposed method yield lower calibration error compared to other methods (that do not use post-training calibration). In terms of raw quality, the proposed model perform on par with the existing method (and frequently slightly better).



**Summary Of The Review:**

- The paper is well executed, but the task of reducing calibration error without post processing is quite narrow and has low impact. The approach of jointly training classification and MLM loss is not novel, but the impact on reducing calibration error is somewhat novel.

---

> ### Author Response · Authors · 2022-11-16
> **Response to Reviewer Kwo1 (Part 1/3)**
>
> Thank you for the careful and valuable review! We address and clarify the questions and concerns below.
>
> Q1: Why do we need a new training paradigm for overcoming lightweight post-processing techniques, such as temperature scaling?
>
> A1: Thanks for pointing this out! The pros and cons of using the common temperature scaling (ts) technique to calibrate the fine-tuned LM have been well discussed in previous work [1]. The empirical study in [1] shows that the PLMs with vanilla fine-tuning have a fundamentally good ECE score in the ID settings, which only need minor rescaling to achieve near-perfect calibration. However, in the OD settings, the effect of ts is limited, and the result ECE scores are far from satisfaction. Here is a brief sketch of the effect of ts under both ID and OD settings which are presented in [1]:
>
> | RoBERTa-base    | ID ECE (No TS-> TS)     | OD ECE (No TS -> TS)      |
> | --------------- | ----------------------- | ------------------------- |
> | SNLI/MNLI       | 1.93 -> 0.84, temp=1.16 | 3.62 -> 1.46, temp=1.25   |
> | QQP/TwitterPPDB | 2.33 -> 0.88, temp=1.39 | 9.55 -> 7.86, temp=2.79   |
> | SWAG/HellaSWAG  | 1.76 -> 0.76, temp=1.10 | 11.93 -> 11.22, temp=2.77 |
>
> Table 1. The effect of temperature scaling under both ID and OD settings on NLI, PD, and CR tasks. The best temperatures in ID and OD settings are selected using ID and OD validation sets, respectively.
>
> From Table 1., we can observe that despite using relatively larger temperature values that are learned on the OD validation set, the impact on ECE under the OD setting is limited, which implies the flaws of the hidden representations given by the fine-tuned LM under large domain shift. Hence, the temperature scaling technique is not sufficient for calibrating the fine-tuned LM in the OD setting. Besides, we show the average predictive confidence of the fine-tuned LM for the outlier samples increases to nearly 100% after 3 training epochs, which can not be fixed by post-processing techniques.
>
> These observations suggest that stronger regularization techniques are needed to improve the representation of OD samples, which can lead to better calibration under OD settings. The previous state-of-the-art method [2] also addresses these and introduces a strong regularization with MixUp to calibrate the LMs. Nevertheless, there needs to be more discussion of why lightweight calibration methods are insufficient for calibrating the fine-tuned LMs in our current version, and we added the discussion in the revised version.
>
> Q2: How the forgetting of pre-trained features by fine-tuned models affects the calibration of downstream tasks is unclear.
>
> A2: We have added some new empirical results to better illustrate the effect of fine-tuning on the model's calibration. In a nutshell, we show that the calibration of the MLM task of the PLM is robust to larger corruption levels than the pre-training phase, which suggests the pre-trained features can model the predictive confidence well under large domain shift. We then show that the fine-tuned model does not preserve this advantage since it projects the inputs to a much simpler data manifold, which makes the fine-tuned model exhibit overconfidence to the OD samples and the outlier samples that do not hold any attributes of the downstream task. We reorganize section 4 (sec 3 in the revised version) based on these new experimental results.
>
>
> Besides, we removed some trivial and non-surprising results, such as Table 1 showing that fine-tuned model can not do MLM.
>
>
> [1] [Calibration of Pre-trained Transformers](https://aclanthology.org/2020.emnlp-main.21) (Desai & Durrett, EMNLP 2020)
>
> [2] [On the Calibration of Pre-trained Language Models using Mixup Guided by Area Under the Margin and Saliency](https://aclanthology.org/2022.acl-long.368) (Park & Caragea, ACL 2022)

---

> > ### Author Response · Authors · 2022-11-16
> > **Response to Reviewer Kwo1 (Part 2/3)**
> >
> > Q3: Using MLM loss in addition to the standard classification loss is known to produce better results. Joint training is also a known technique.
> >
> > A3: Indeed, the idea of incorporating the MLM loss with standard classification loss itself is not novel. As you have commented, the novelty here is that using the MLM loss during the fine-tuning process to better calibrate the fine-tuned LM, while previous works focus on improving the raw quality. However, we agree that the presentation of the current version place too much emphasis on the joint learning method (e.g., the title and "a new fine-tuning criterion" in sec. 5), which outweighs the key idea of our work.  The proposed method should be treated as a validation of our hypothesis that preserving the pre-trained features of PLMs is helpful for the calibration of the fine-tuned LMs on downstream tasks. We adjusted the presentations about the proposed joint training with the MLM loss method in the revised version.
> >
> > Furthermore, we decide to take the constructive advice from you and reviewer eTyr to discuss the methods from the catastrophic forgetting literature and add two new baselines: weight decay towards the pre-trained weight and Mixout [3].
> >
> >
> > Q4: The scope/impact of this work is quite narrow.
> >
> > A4: First, we'd like to emphasize that our paper's main contribution is not the proposed method itself but the new perspective and findings for the calibration of fine-tuned LMs. Unlike previous general-purpose calibration methods (TS, mixup, etc.), our work focuses specifically on the pre-training & fine-tuning paradigm. We reveal a previously omitted fact that **we actually have an amazingly calibrated model**, the PLM, due to its large and broad training data. We further show that the capacity of the PLM for making well-calibrated predictions **can be transferred to downstream tasks simply by preserving the pretraining features**. We argue that these findings open a new avenue for improving the calibration of fine-tuned models by leveraging the PLMs and, subsequently, the massive amount of pre-training corpus. Our adopted methods, such as fine-tuning with the MLM objective, are indeed simple. However, the positive results of these simple methods further strengthen the potential of our new perspective, and the result might be improved by more sophisticated future research.
> >
> > Besides, as addressed in A1, in order to better calibrate the fine-tuned LMs, especially under the OD settings, we need strong regularization methods to fix the flaws of the fine-tuned language model in confidence modeling. We show that preserving the pre-trained features during fine-tuning is an effective way to calibrate the fine-tuned LMs while having raw quality on par, which is competitive with other calibration methods. Furthermore, such training paradigms that can mitigate catastrophic forgetting have other promising advantages, such as better classification performance [3], OOD generalization [4], and adversarial robustness [5].
> >
> > [3] [Mixout: Effective Regularization to Finetune Large-scale Pretrained Language Models](https://arxiv.org/abs/1909.11299) (Lee, Cheolhyoung, Kyunghyun Cho, and Wanmo Kang, ICLR 2020)
> >
> > [4] [Composed Fine-Tuning: Freezing Pre-Trained Denoising Autoencoders for Improved Generalization](https://proceedings.mlr.press/v139/xie21f) (Xie et al., ICML 2021)
> >
> > [5] [How Should Pre-Trained Language Models Be Fine-Tuned Towards Adversarial Robustness?](https://proceedings.neurips.cc/paper/2021/file/22b1f2e0983160db6f7bb9f62f4dbb39-Paper.pdf) (Dong, Xinshuai, et al., NIPS 2021)

---

> > > ### Author Response · Authors · 2022-11-16
> > > **Response to Reviewer Kwo1 (Part 3/3)**
> > >
> > > Q5: Unfair comparsion to the baseline since additional training data was used.
> > >
> > > A5: The introduced unsupervised training data for the MLM task is an additional cost compared to the baselines, and we will address this limitation in the updated version. Besides, in our revised version, we show that among several baselines (parameter-efficient tuning, Weight decay to pre-trained weight, and Mixout),  performing the MLM task on additional pre-training corpus with knowledge distillation is the most effective way to preserve the pre-trained features that are helpful to the calibration while yielding competitive raw quality. Based on the difference between the baseline methods, we discuss the room for future work to improve the existing methods further.
> > >
> > > Q6: The training time of the joint learning method is 2x longer.
> > >
> > > A6: This is a major drawback of the joint learning method. The training scheme causes 1.5x-2x training time (in some criteria that use large batch size, the gap of total time cost between vanilla fine-tuning and joint learning can be narrowed).  We clarify this and add the comparison of the training time between different methods in the appendix of the updated version.
> > >
> > > Q7: About Sec 5.1, the DVAE framework lost the connection with the rest of the paper.
> > >
> > > A7: As addressed in A3, we decided to take your and reviewer eTyr's advice and drop the DVAE framework and treat the joint learning methods as a baseline that preserves the pre-trained features along with other methods.

---

> > > > ### Comment · Reviewer_Kwo1 · 2022-11-24
> > > > **Response**
> > > >
> > > > The authors did a serious job in addressing the existing concerns, including comparison to temperature scaling, phrasing the paper as studying the impact of catastrophic forgetting on calibration, and fairly discussing the cost impact of additional MLM training.
> > > >
> > > > The updated manuscript will be a nice contribution to our understanding of calibration and to the ICLR community.

---

> > > > > ### Author Response · Authors · 2022-11-24
> > > > > **Thanks!**
> > > > >
> > > > > It's great to see that your concerns are appropriately addressed and acknowledge our contribution. Thank you again for your efforts and valuable reviews!

---

### Author Response · Authors · 2022-11-16
**Response to all reviewers and ACs & Summary of the Revisions**

We thank all reviewers and ACs for your effort and devotion. We revise and reframe our work based on your valuable comments and constructive advice. Here is a summary of the revisions:

* REVISE THE TITLE to Preserving Pre-trained Features Helps Calibrate Fine-tuned Language Models.
* Strengthen the discussion on the significance and motivation for the problem of calibrating fine-tuned LMs.
* Address why lightweight post-hoc calibration methods are not sufficient for calibrating the fine-tuned LMs.
* Move the related work section to the end and address the contributions and impacts of our work based on previous works.
* Remove some trivial results, e.g., fine-tuned LMs are not able to perform MLM.
* Move the discussion about parameter-efficient tuning to the method section.
* Add experimental results and discussion about calibration of MLM task under different corruption levels.
* Add experimental results and discussion about how the hidden representation for cross-domain samples changes after fine-tuning.
* Add experimental results and discussion about the fine-tuned LM's predictive confidence for out-of-domain and outlier samples during fine-tuning process.
* Drop the DVAE framework and reframe the method section.
* Adjust the motivation for adopting the knowledge distillation.
* Add two more baselines that preserve the pre-trained features.
* Clarify the limitation of additional dataset usage and longer training time of the proposed joint learning methods
* Add experimental results and discussion about how the existing/proposed methods that preserve the pre-trained features affect the model's predictive confidence.
* Move the ablation study and sampling results of the joint learning with MLM loss methods to the appendix.
* Add experimental results of confidence histogram on the ablation study part.

We also address the question and concerns of the reviewers in detail and would like to clearify any further concerns. We are open to further improve our work in all aspects.

---

### Author Response · Authors · 2022-11-21
**Looking forward to further discussions!**

Dear reviewers,

In the ended discussion stage 1, we revised and reframed our paper based on your valuable and constructive reviews. With the original motivation and observations, we adjusted the presentation to focus on our key idea that preserving the pre-trained features helps calibrate the fine-tuned LMs and added more experimental results to better illustrate and validate the key idea. We would appreciate it if you could check our responses and re-evaluate our paper.

We look forward to discussing any further questions/concerns/suggestions with you in the discussion stage 2.

Again, thanks for your time and effort in reviewing our paper!

Best,

Authors of #6210

---

### Decision · Program_Chairs · 2023-01-20

**Decision:**

Accept: poster

**Justification For Why Not Higher Score:**

1) The impact is limited.
2) The revised version is quite different from the original lowering the reviewers confidence

**Justification For Why Not Lower Score:**

The paper provides a thorough analysis to an interesting problem, its easy to read and would be interesting to the ML community, as well as motivate follow up works providing better solutions to the highlighted problem

**Metareview: Summary, Strengths And Weaknesses:**

The paper addresses a problem where fine-tuned pre-trained language models are not well calibrated. They observe that the problem lies in the fine-tuning phase and suggest to rectify this by modifying the fine-tuning process in a way that forces it to preserve (in some way) the pre-trained weights. The paper does not suggest a new method for fine tuning, rather explores several known methods in terms of their performance and calibration, both in distribution and out of distribution.

In its original version, the paper suggested a solution involving Denoising Variational Auto-Encoders. The reviews mentioned this to be an unnecessary addition that only confuses the reader. As a result, the authors rewrote the paper (and changed its title) making it clear and easy to read, as well as an honest presentation of the real contribution: Not a new method but a study of existing methods.

In the discussion phase I found that the scores for the paper remain only mildly enthusiastic due to two reasons: (1) the impact of the paper may be considered limited as it doesn't provide a new method, but a study of existing methods in a new lens (of calibration), (2) the changes are substantial, and some of the reviewers did not have the time to properly review the revised version.

As for (1), given the overall support from all 3 reviews as well as my opinion, I believe that studying the issue of calibration is well motivated and a thorough study should be of interest to the ICLR community. For issue (2), I went over the revised version and find it easy to read, and agree with the reviewers that the main issues of clarity w.r.t other methods and motivation are solved.





**Note From Pc:**

if the above contains the word "oral" or "spotlight" please see: "oral" presentation means -> notable-top-5% and "spotlight" means -> notable-top-25%. As stated in our emails, we are disassociating presentation type from AC recommendations

**Summary Of Ac-Reviewer Meeting:**

n/a